# The Effect of Physical Activity and High Body Mass Index on Health-Related Quality of Life in Individuals with Metabolic Syndrome

**DOI:** 10.3390/ijerph17103728

**Published:** 2020-05-25

**Authors:** Alba Marcos-Delgado, Tania Fernández-Villa, Miguel Ángel Martínez-González, Jordi Salas-Salvadó, Dolores Corella, Olga Castañer, J. Alfredo Martínez, Ángel M. Alonso-Gómez, Julia Wärnberg, Jesús Vioque, Dora Romaguera, José López-Miranda, Ramon Estruch, Francisco J Tinahones, José Lapetra, J. LLuís Serra-Majem, Laura García-Molina, Josep A. Tur, José Antonio de Paz, Xavier Pintó, Miguel Delgado-Rodríguez, Pilar Matía-Martín, Josep Vidal, Clotilde Vázquez, Lidia Daimiel, Emilio Ros, Nancy Babio, Ignacio M Gimenez-Alba, Estefanía Toledo, María Dolores Zomeño, M. A. Zulet, Jessica Vaquero-Luna, Jessica Pérez-López, Ana Pastor-Morel, Aina M Galmes-Panades, Antonio García-Rios, Rosa Casas, María Rosa Bernal-López, José Manuel Santos-Lozano, Nerea Becerra-Tomás, Carolina Ortega-Azorin, Zenaida Vázquez-Ruiz, Karla Alejandra Pérez-Vega, Itziar Abete, Carolina Sorto-Sánchez, Antoni Palau-Galindo, Iñigo Galilea-Zabalza, Júlia Muñoz-Martínez, Vicente Martín

**Affiliations:** 1Institute of Biomedicine (IBIOMED), University of León, 24071 León, Spain; amarcd@unileon.es (A.M.-D.); japazf@unileon.es (J.A.d.P.); vmars@unileon.es (V.M.); 2Centro de Investigación Biomédica en Red Fisiopatología de la Obesidad y la Nutrición (CIBEROBN), Institute of Health Carlos III, 28040 Madrid, Spain; mamartinez@unav.es (M.Á.M.-G.); jordi.salas@urv.cat (J.S.-S.); dolores.corella@uv.es (D.C.); ocastaner@imim.es (O.C.); jalfmtz@unav.es (J.A.M.); angelmago13@gmail.com (Á.M.A.-G.); jwarnberg@uma.es (J.W.); mariaadoracion.romaguera@ssib.es (D.R.); jlopezmir@gmail.com (J.L.-M.); restruch@clinic.cat (R.E.); fjtinahones@uma.es (F.J.T.); joselapetra543@gmail.com (J.L.); lluis.serra@ulpgc.es (J.L.S.-M.); pep.tur@uib.es (J.A.T.); xpinto@bellvitgehospital.cat (X.P.); cvazquezma@gmail.com (C.V.); eros@clinic.cat (E.R.); nancy.babio@urv.cat (N.B.); i.gimenez.alba@valencia.edu (I.M.G.-A.); etoledo@unav.es (E.T.); jessicaperezlopez@uma.es (J.P.-L.); aina.galmes.panades@gmail.com (A.M.G.-P.); angarios2004@yahoo.es (A.G.-R.); rcasasr@gmail.com (R.C.); robelopajiju@yahoo.es (M.R.B.-L.); jsantos11@us.es (J.M.S.-L.); nerea.becerra@urv.cat (N.B.-T.); carolina.ortega@uv.es (C.O.-A.); zvazquez@unav.es (Z.V.-R.); apalau@grupsagessa.com (A.P.-G.); inigo.galilea.zabalza@navarra.es (I.G.-Z.); 3Department of Preventive Medicine and Public Health, IdiSNA, University of Navarra, 31008 Pamplona, Spain; 4Department of Nutrition, Harvard T.H. Chan School of Public Health, Boston, MA 02115, USA; 5Departament de Bioquímica i Biotecnologia, Unitat de Nutrició, Universitat Rovira i Virgili, 43201 Reus, Spain; 6Nutrition Unit, University Hospital of Sant Joan de Reus, 43204 Reus, Spain; 7Institut d’Investigació Sanitària Pere Virgili (IISPV), 43204 Reus, Spain; 8Department of Preventive Medicine, University of Valencia, 46010 Valencia, Spain; 9Unit of Cardiovascular Risk and Nutrition, Institut Hospital del Mar de Investigaciones Médicas Municipal d’Investigació Médica (IMIM), 08003 Barcelona, Spain; mzomeno@imim.es (M.D.Z.); kperez@imim.es (K.A.P.-V.); jmunoz@imim.es (J.M.-M.); 10Department of Nutrition, Food Sciences, and Physiology, Center for Nutrition Research, University of Navarra, 31009 Pamplona, Spain; mazulet@unav.es (M.A.Z.); iabetego@unav.es (I.A.); 11Precision Nutrition and Cardiometabolic Health Program, IMDEA Food, CEI UAM + CSIC, 28049 Madrid, Spain; 12Bioaraba Health Research Institute, Cardiovascular, Respiratory and Metabolic Area, Osakidetza Basque Health Service, Araba University Hospital, University of the Basque Country UPV/EHU, 48940 Vitoria-Gasteiz, Spain; luna_jess_@hotmail.com (J.V.-L.); daisysorto2@yahoo.com (C.S.-S.); 13Department of Nursing, Institute of Biomedical Research in Malaga (IBIMA), University of Málaga, 29010 Málaga, Spain; 14CIBER de Epidemiología y Salud Pública (CIBERESP), Instituto de Salud Carlos III, 28029 Madrid, Spain; vioque@umh.es (J.V.); lagarmol1@gmail.com (L.G.-M.); 15Instituto de Investigación Sanitaria y Biomédica de Alicante, ISABIAL, 03010 Alicante, Spain; 16Health Research Institute of the Balearic Islands (IdISBa), University Hospital Son Espases, 07120 Palma de Mallorca, Spain; 17Department of Internal Medicine, Maimonides Biomedical Research Institute of Cordoba (IMIBIC), Reina Sofia University Hospital, University of Cordoba, 14004 Cordoba, Spain; 18Department of Internal Medicine, Institut d’Investigacions Biomèdiques August Pi Sunyer (IDIBAPS), Hospital Clinic, University of Barcelona, 08036 Barcelona, Spain; 19Virgen de la Victoria Hospital, Department of Endocrinology, Instituto de Investigación Biomédica de Málaga (IBIMA), University of Málaga, 29010 Málaga, Spain; 20Department of Family Medicine, Research Unit, Distrito Sanitario Atención Primaria Sevilla, 29009 Sevilla, Spain; 21Research Institute of Biomedical and Health Sciences (IUIBS), University of Las Palmas de Gran Canaria & Centro Hospitalario Universitario Insular Materno Infantil (CHUIMI), Canarian Health Service, 35016 Las Palmas de Gran Canaria, Spain; 22Department of Preventive Medicine and Public Health, University of Granada, 18010 Granada, Spain; 23Research Group on Community Nutrition & Oxidative Stress, University of Balearic Islands, 07122 Palma de Mallorca, Spain; 24Lipids and Vascular Risk Unit, Internal Medicine, Hospital Universitario de Bellvitge, Hospitalet de Llobregat, 08907 Barcelona, Spain; 25Nutritional Genomics and Epigenomics Group, IMDEA Food, CEI UAM + CSIC, 28049 Madrid, Spain; mdelgado@ujaen.es; 26Division of Preventive Medicine, Faculty of Medicine, University of Jaén, 23071 Jaén, Spain; 27Department of Endocrinology and Nutrition, Instituto de Investigación Sanitaria Hospital Clínico San Carlos (IdISSC), 28040 Madrid, Spain; mmatia@ucm.es; 28CIBER Diabetes y Enfermedades Metabólicas (CIBERDEM), Instituto de Salud Carlos III (ISCIII), 28029 Madrid, Spain; jovidal@clinic.cat; 29Department of Endocrinology, Institut d’Investigacions Biomédiques August Pi Sunyer (IDIBAPS), Hospital Clinic, University of Barcelona, 08036 Barcelona, Spain; 30Department of Endocrinology and Nutrition, Hospital Fundación Jimenez Díaz, Instituto de Investigaciones Biomédicas IISFJD, University Autonoma, 28040 Madrid, Spain; 31Nutritional Control of the Epigenome Group, Precision Nutrition and Obesity Program, IMDEA Food, CEI UAM + CSIC, 28049 Madrid, Spain; lidia.daimiel@imdea.org; 32Lipid Clinic, Department of Endocrinology and Nutrition, Institut d’Investigacions Biomèdiques August Pi Sunyer (IDIBAPS), Hospital Clínic, 08036 Barcelona, Spain; 33Centro de Salud Cabo Huertas, 03540 Alicante, Spain; apastormorel@gmail.com; 34Departament of Internal Medicine, Regional University Hospital of Malaga, Instituto de Investigación Biomédica de Málaga (IBIMA), 29010 Málaga, Spain

**Keywords:** health-related quality of life, physical activity, obesity, body mass index, metabolic syndrome

## Abstract

The main objective of this study was to examine the relationship between the level of physical activity (PA) and the degree of obesity with health-related quality of life (HRQoL) in individuals with metabolic syndrome (MetS) who participated in the Predimed-Plus study. A total of 6875 subjects between 55 and 75 years of age with MetS were selected and randomized in 23 Spanish centers. Subjects were classified according to categories of body mass index (BMI). PA was measured with the validated Registre Gironí del Cor (REGICOR) questionnaire and subjects were classified according to their PA level (light, moderate, vigorous) and the HRQoL was measured with the validated short-form 36 (SF-36) questionnaire. By using the ANOVA model, we found a positive and statistically significant association between the level of PA and the HRQoL (aggregated physical and mental dimensions *p* < 0.001), but a negative association with higher BMI in aggregated physical dimensions *p* < 0.001. Furthermore, women obtained lower scores compared with men, more five points in all fields of SF-36. Therefore, it is essential to promote PA and body weight control from primary care consultations to improve HRQoL, paying special attention to the differences that sex incurs.

## 1. Introduction

According to the World Obesity Federation, obesity is a chronic, recurring disease, considered to be the epidemic of the 21st century, due to its increase in recent decades [1]. More than 1.5 trillion adults worldwide are overweight and 500 million of who suffer from obesity [2,3]. These figures have doubled since 1980 and currently almost three million deaths per year are attributed to excess weight, a trend that indicates overweight and obesity will cause more deaths worldwide than malnutrition [4,5].

Overweight/obesity and excess visceral fat in particular, represent a higher risk of morbidity and mortality [6,7]. It is also associated with diseases like type two diabetes, high blood pressure, dyslipidemia, cardiovascular diseases, sleep apnea [7,8,9], some types of cancer [10] and increased social stigma [11]. In addition to diseases being associated with obesity, a major obstacle to physical activity for adults with obesity, are impairments in movement and walking [12,13]. The epidemiological model of obesity describes diet as the main causative agent of excess weight and scarce physical activity (PA) as the second main driver [1,14]. Fast food, including consumption of ultraproccessed foods, is currently replacing traditional diet and the percentage of people who exercise regularly in their spare time is continuously decreasing and becomes even less frequent with age. In Spain, 12.4% of people between 45%–64% and 5.9% of people 65 years and older [15] exercise regularly. As such, obesity and lack of PA seem to have a negative effect on health-related quality of life (HRQoL) [16,17].

According to the World Health Organization’s (WHO) definition, we can associate HRQoL with the impact a disease has on our perception of happiness, position in life and on our physical, psychological, social and spiritual well-being [18]. Evidence suggests that doing 150 to 300 min of moderate PA per week and following the Mediterranean diet provide considerable benefits in maintaining health, healing and in HRQoL [17,19,20,21,22,23,24]. Overweight/obesity are the main determinants of metabolic syndrome (MetS) and there are increasing numbers of studies that use HRQoL in individuals with MetS, since it is considered a strong predictor of disability and long-term mortality [25,26,27].

Therefore, the main objective of our study is to evaluate whether higher body mass index (BMI) and doing PA are cross-sectionally associated with a better HRQoL in the participants of the multicenter randomized primary prevention trial Predimed-Plus. The final objective is to be able to incorporate strategies that improve the population’s HRQoL, since professionals at the primary care level are in a position to stimulate, reinforce, teach and facilitate healthy habits in patients.

## 2. Materials and Methods

### 2.1. Study Design

A cross-sectional descriptive analysis was carried out with the participants in the Predimed-Plus study, a multicenter randomized trial, in which a total of 23 Spanish centers participated. The aim was to evaluate the effect of an intensive intervention with a hypocaloric Mediterranean diet associated with physical exercise and behavioral therapy compared to a control group that received a traditional Mediterranean diet for primary prevention of cardiovascular diseases in Spain. The protocol can be found at https://www.predimedplus.com/ [28] and the study design was described elsewhere [29,30].

### 2.2. Ethical Considerations

This trial was approved by the Institutional Review Boards of all recruitment centers where the study was conducted [29]. The investigations were carried out following the rules of the Declaration of Helsinki of 1975, revised in 2013.

Participants signed a written informed consent. The trial was registered in 2014 at the International Standard Randomized Controlled Trial (ISRCTN89898870).

### 2.3. Participants

The inclusion criteria of the participants in the Predimed-Plus study were men between the ages of 55–75 years and women between 60–75 years, overweight or obese (BMI ≥ 27 and <40 kg/m^2^), who had at least three criteria of MetS [31] and did not have cardiovascular disease. A total of 6874 subjects were recruited and randomized. For the current analysis, we excluded 130 who did not answered REGICOR questionnaire for PA and 24 participants who did not have available data for BMI at the time. A total of 6720 participants were included for evaluation in the current study. The data in this analysis were collected before the intervention.

### 2.4. HRQoL

The HRQoL was measured with the Spanish version of the SF-36 questionnaire [32,33], validated for the Spanish population and widely used as an accurate way to measure self-perceived HRQoL. This questionnaire consisted of 36 items that assessed eight dimensions or scales: physical functioning (PF), role—physical (RP), bodily pain (BP), general health (GH), vitality (VT), social functioning (SF), role—emotional (RE) and mental health (MH). These dimensions were grouped into two health components: the physical component summary (PCS) and the mental component summary (MCS).

Cronbach’s α was used to measure the reliability of the SF-36 scales and found that the values for the Spanish population aged ≥60 years was higher than the proposed standard of 0.7.

Each item received a numerical score that was encoded, summed up and put on a scale from 0 to 100. The higher the score, the better the quality of life in the analyzed field.

### 2.5. PA and BMI

PA was measured with a short version of the REGICOR questionnaire, validated for the Spanish population [34]. It included the type of PA, how often it was done in the past month (number of days) and the duration of each session, in minutes. The intensity of each PA was measured with metabolic equivalents (METs), according to the Compendium of Physical Activities [35]. The total energy expenditure was calculated by multiplying the METs assigned to each activity by the number of times done per months, minutes per day and divided by 4 weeks/month (METs. min/week) to classify PA as light, moderate or vigorous.

Weight and height were collected the same way in all the centers, by trained personnel following a pre-established protocol. BMI was calculated by dividing the weight (Kg) by the squared height (m^2^). Lastly, the sample population was categorized into four groups (BMI < 30 kg/m^2^, between 30 kg/m^2^ and 32.4 kg/m^2^, between 32.5 kg/m^2^ and 34.9 kg/m^2^ and ≥35 kg/m^2^).

### 2.6. Other Covariables

The sociodemographic variables were collected with the general questionnaire administered at the first visit were sex, age, level of education (primary, secondary and higher education) and marital status (single, married, divorced/separated and widowed).

Other covariates used for adjusted models were measurements for high blood pressure, smoking (non-smoker, smoker, ex-smoker), type 2 diabetes, depression, chronic obstructive pulmonary disease and previous history of cancer.

### 2.7. Statistical Analysis

Participants were divided in three categories of PA based on the intensity measured with the short version of the REGICOR questionnaire: Light PA (<1200 METs–min/week), moderate (1200–2700 METs–min/week) and vigorous (>2700 METs–min/week).

The BMI variable was categorized in four categories: <30 kg/m^2^, between 30 kg/m^2^ and 32.4 kg/m^2^, between 32.5 kg/m^2^ and 34.9 kg/m^2^ and ≥35 kg/m^2^.

In the sample description, means and standard deviations for quantitative variables were calculated, as well as the percentages for qualitative variables, consistent with the defined categories for PA and BMI. PA levels were compared with age and BMI with ANOVA test and with type 2 diabetes, depression, blood pressure, history of cancer, history of lung disease, smoking status, marital status and educational level with a χ^2^ test.

The mean of each dimension of the SF-36 questionnaire and of the two additional fields in the three PA categories and BMI were compared with ANOVA models. The differences between groups were calculated using linear regression models, stratified by sex. Three adjustment models were done for each scale of the HRQoL questionnaire. The first model was adjusted for age and educational level (primary, secondary and higher education). In the second PA model, BMI (<30 kg/m^2^, between 30 kg/m^2^ and 32.4 kg/m^2^, between 32.5 kg/m^2^ and 34.9 kg/m^2^ and ≥35 kg/m^2^), smoking habit (non-smoker, smoker, ex-smoker) and marital status (single, married, divorced/separated, widowed) were included and, with BMI as a dependent variable, PA (light, moderate and vigorous) was included in the model as an adjustment variable. Lastly, the third model was adjusted for comorbidities that are associated with lower HRQoL, high blood pressure (yes/no), type 2 diabetes (yes/no), depression (yes/no), chronic obstructive pulmonary disease (yes/no) and previous history of cancer (yes/no). Benjamini-Hochberg Procedure (BH) was used as a correction test.

Linear trends for the association of PA, BMI and HRQoL dimensions were also calculated. For this, we considered the PA and BMI categories to be a continuous quantitative variable included in the multivariate adjusted models. The data were analyzed using the available, complete PREDIMED-Plus database, dated 25/03/2019.

All statistical analyses were done using the StataCorp statistical package. 2015. *Stata Statistical Software: Release 15.* College Station, TX, USA: StataCorp LP. We considered a two-tailed value of 0.05 to be the threshold for statistical significance.

## 3. Results

The descriptive characteristics of the sample in terms of PA levels and BMI are shown in Table 1 and Table 2. A decrease in the prevalence in the pathologies studied was seen with increasing intensity of PA. Conversely, in the BMI categories, the lower the BMI, the lower the prevalence of chronic conditions such as type 2 diabetes, depression, high blood pressure and a history of lung disease. Furthermore, the prevalence of pathologies like depression and a history of cancer were higher in women and a higher percentage of women are smokers and have primary education.

Appendix A show the adjusted means of the eight dimensions that make up the HRQoL SF-36 questionnaire in the three PA categories. As shown, the higher the level of PA, the higher the average scores in HRQoL and, furthermore, all scores were lower in women than in men (Figure 1 and Figure 2). Using the third adjustment model as a reference, we observed statistically significant differences on all scales in women (*p* < 0.001), while in men no differences were found in role—physical (*p* = 0.1686), bodily pain (*p* = 0.4762) and role—emotional (*p* = 0.0756).

Within the scales that make up the PCS, we observed a difference of ten points in the means between light activity, (PF 65.14 (95% CI 64.07–66.22) RP 61.73 (95% CI 59.59–63.87)); and vigorous activity (PF 75.27 (95% CI 74.00–76.54) RP 73.37 (95% CI 70.84–75.91)) in the PF and RP scales in women. On the BP and GH scales, the difference is eight points. In men, there is a five-point difference between the levels of PA in the PF and GH scales and a two-point difference in RF and BP scales.

In the MCS component there were significant differences (*p* < 0.0001) in all scales between the average scores found in terms of the PA category in women and men (the higher the level of physical exercise, the higher the HRQoL scores) except in the RE scale in men (*p* = 0.092). In women there was a difference of ten points in all fields except MH, where the difference was nine points. The greatest difference in men was found in the VT field, where VT was seven points over the PA average, in SF and RE, the difference was three points and in MH there was a difference of four points.

Appendix A show the average scores obtained in the eight dimensions of the SF-36 questionnaire in relation to the BMI categories. In the PCS component, the higher the BMI, the lower the average score of the dimensions in both sexes. All scores were lower in women than in men (Figure 3 and Figure 4). The fields with a greater difference in both sexes were PF and BP, with a difference of seven and ten points, respectively.

In terms of the relationship between MCS and BMI, for women, the higher the degreed of obesity, the lower the average scores on the VT (*p* < 0.001) and SF (*p* = 0.026) scales, while in RE and MH, no statistically significant differences were observed. In men, the average scores tended to increase slightly as the BMI increased, except in the vitality field where there was a significant decreased between the lowest and highest BMI category (*p* = 0.0001). The greatest change in both sexes was in the VT field, with a difference of four points in women and three points in men.

## 4. Discussion

In our study, there was a significant increase in the scores for women in all dimensions that making up the HRQoL SF-36 questionnaire when more PA was done. There was also a significant increase for men in all fields except RP, BP and RE. Regarding the association between HRQoL and BMI, we found an inversely proportional relationship in the areas that make up PCS in both women and men. Furthermore, we could observe that this association is also significant in the mental component in women.

This suggests that both PA and obesity/overweight were determinants in HRQoL. We can also saw that all scores were lower in women than in men, which was consistent with previous published articles [22,36,37].

Jantunen et al. [38] assessed the association between PA and HRQoL for ten years in a Finnish cohort, where a statistically significant association was observed between increased PA and improving the physical component dimension in men and women. A statistically significant association was also found in women for the mental component dimension, which was similar to our study. Other research suggested that there were benefits of PA to improve physical and mental state, increase physical function, reduce anxiety, depression and stress through various mechanisms, like the effect of endorphins, thermogenic hypothesis or the effect of the immune system through cytokines, among others [39,40].

As such and in relation to other previous published studies, the same trend occurred with obesity and HRQoL [41,42,43,44]. In the systematic review of reviews conducted by Kolotkin et al. [16], articles included, that were similar to our study verified that the scores in the physical component field were lower when the BMI ≥ 25 kg/m^2^, a relationship make even more evident in higher BMI categories, like in our subjects. The lowest scored dimensions within the physical component field, for both men and women, were PF and BP. This may be due to the relationship between obesity and chronic pain, including generalized joint pain and other musculoskeletal pain, which in turn interferes with physical functioning. The mechanisms involved are still unclear, however, one of the most accepted theories is that excess weight exerts mechanical pressure on the joints and intervertebral discs, which contributes directly and indirectly to pain. Another related factor seems to be sedentary lifestyle; people with obesity who practice little PA have more lower back pain than people without obesity, which is consistent with our study [45].

In the study by Fanning et al. [22] compared the effect of weight loss alone or in combination with aerobic or resistance training in individuals with MetS on HRQoL, although our study is not comparable since it was a cross-sectional, it was found that there was a significant improvement in physical domain scores on the SF-12 questionnaire in those individuals who combined weight loss with either aerobic or resistance training, which leads us to believe that it is the combination of adequate BMI and PA that will result in the greatest increase in HRQoL.

In terms of the mental field, there was a worsening of HRQoL in women with higher BMI, specifically in the VT and SF dimensions, but this did not occur in men. One of the reasons could be the effect that the stigma of obesity has on women compared to men [46].

By contrast, there were several studies in which the participants’ perceived HRQoL did not change after an intervention with PA. One was a study conducted in Norway were a physical exercise intervention was carried out for 12 weeks on patients who had hip surgeries [47] and the other was a 12-month exercise program with women over 65 years of age [48]. However, when comparing these data, we must take into consideration that the participants in our study were otherwise healthy people with cardiovascular risk factors

On the other hand, it was proven that the type of exercise performed seems to have no influence on HRQoL [49], which leads us to believe that the simple act of doing PA is what improves HRQoL.

Pozas et al. [50] in their study of the association between MetS and HRQoL involving 229 participants, 118 with MetS and 111 without MetS, found no greater worsening of HRQoL in the total sample of individuals with MetS compared to the group without MetS. However, when BMI was analyzed, they found that there was an inverse relationship between BMI and HRQoL in both groups, which may lead to believe that it is the degree of BMI that influences HRQoL and not the MetS. On the other hand, in this same study, unlike ours, it was the men with MetS who obtained significantly worse scores in the RE dimension.

We can saw that all scores were lower in women than in men. This difference could be explained by sociodemographic characteristics like income level, level of education and some chronic diseases, as well as the aforementioned effect that the stigma of high BMI and obesity may exert on some women [46,51].

This study had some limitations that need to the highlighted. First, it was a cross-sectional study, meaning that it did not imply causation. Second, also inherent to its design, we cannot exclude the possible inverse causation, that PA and a normal BMI can improved HRQoL, but also a good HRQoL can encouraged physical exercise and contributed to obtaining an ideal weight. Third, the result cannot be extrapolated to other study populations since our analyzes included only elderly Mediterranean individuals with MetS.

In terms of our study’s strengths, it is important to note that we had a high number of participants and accurate validity and reproducibility to measure the HRQoL of the SF-36 questionnaire.

## 5. Conclusions

In conclusion, our results suggest how lack of PA were negatively associated to aggregated physical and mental dimensions (*p* < 0.001) and obesity were negatively associated to aggregated physical dimensions (*p* < 0.001) that comprise HRQoL. Women suffered the most severe deterioration compared to men, more five points in all fields of SF-36. Therefore, it is essential to promote PA and body weight control from the primary care standpoint through consultations to improve HRQoL, paying special attention to the differences that sex incurs.

## Figures and Tables

**Figure 1 ijerph-17-03728-f001:**
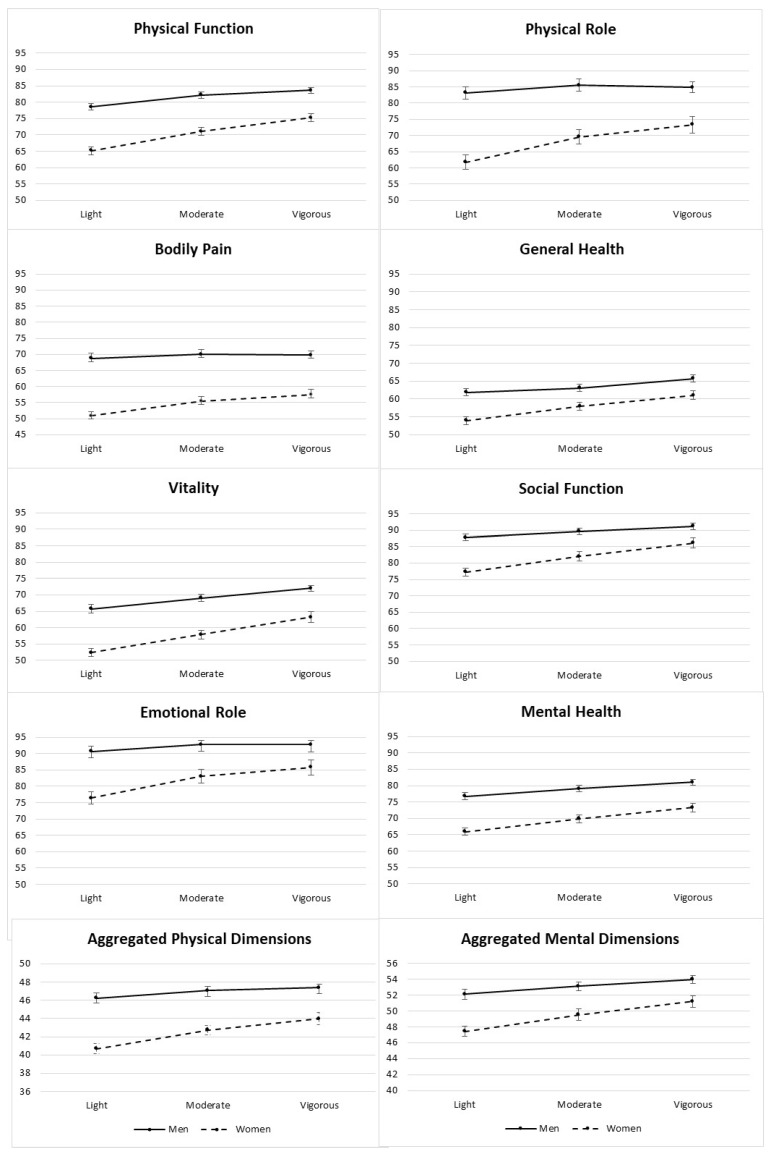
Adjusted means (points) for each of the 8 dimensions and aggregated physical/mental dimensions of health-related quality of life (HRQoL) by baseline categories of PA (metabolic equivalents (METs).min/week).

**Figure 2 ijerph-17-03728-f002:**
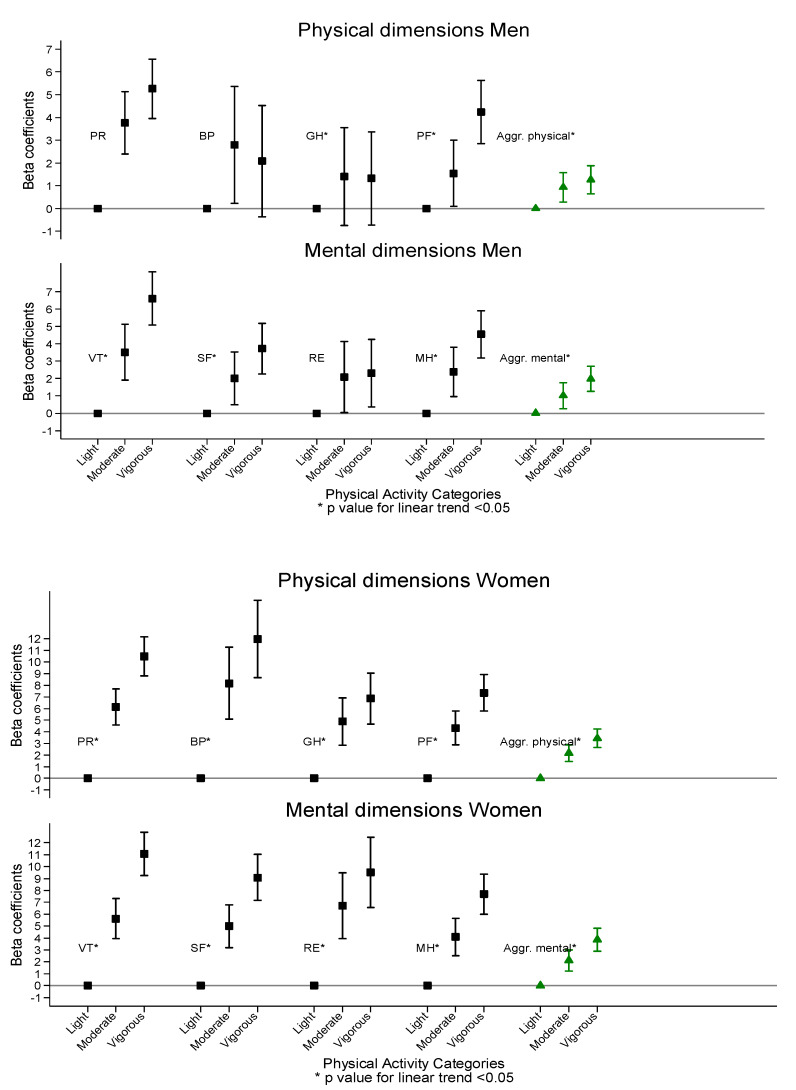
Mean differences and confidence intervals in the HRQoL dimensions according to PA categories in the Predimed-Plus trial.

**Figure 3 ijerph-17-03728-f003:**
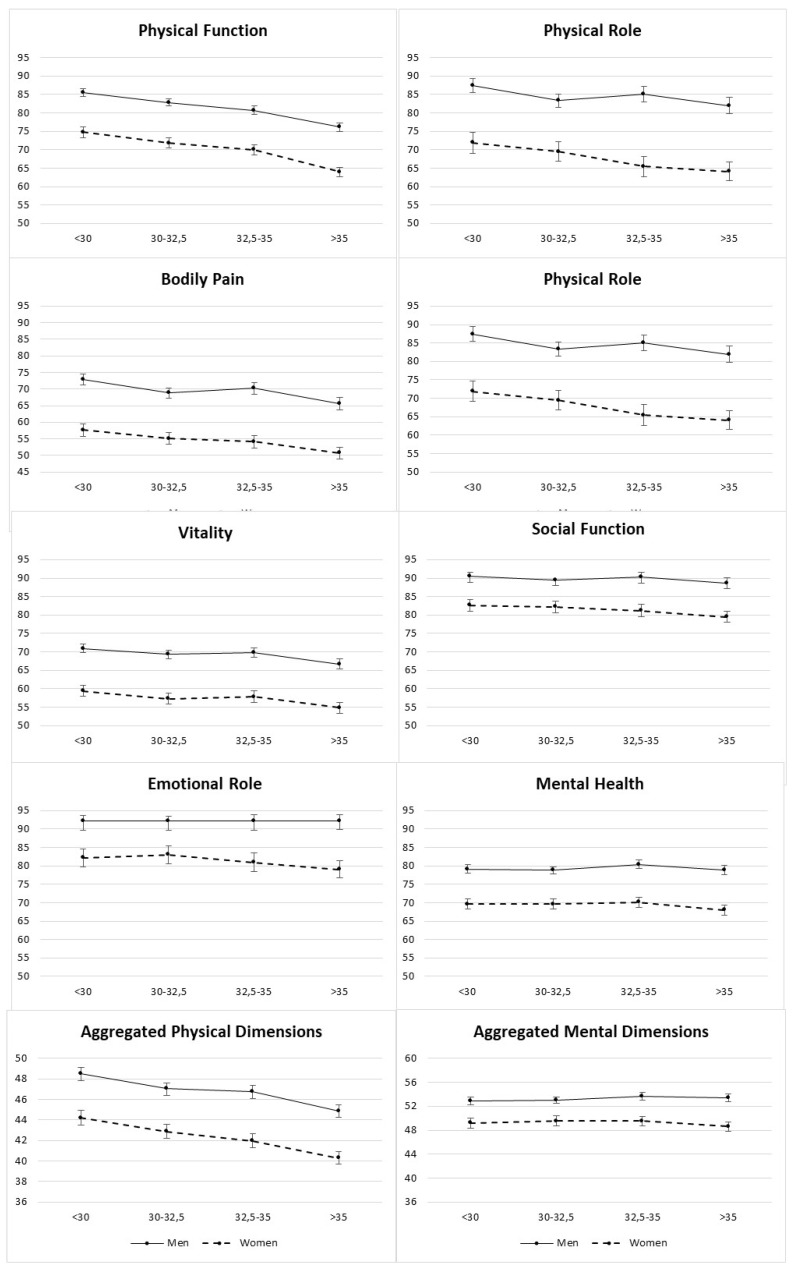
Adjusted means (points) for each of the 8 dimensions and aggregated physical/mental dimensions of HRQoL by baseline categories of BMI (kg/m^2^).

**Figure 4 ijerph-17-03728-f004:**
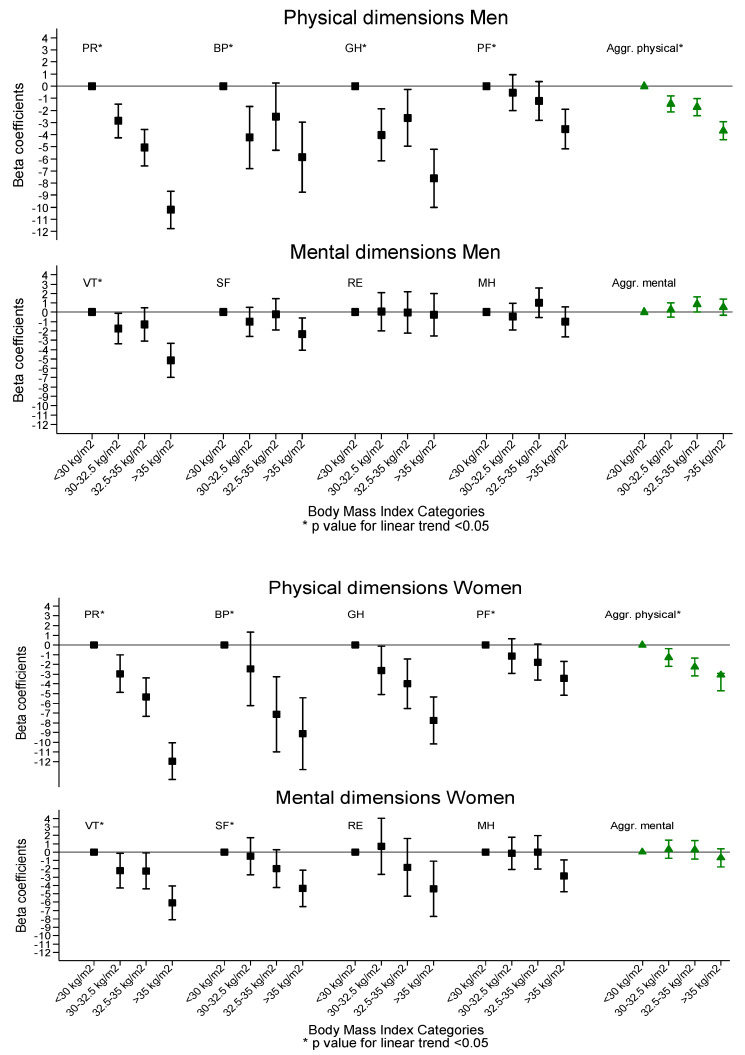
Mean differences and confidence intervals in the HRQoL dimensions according to BMI categories in the Predimed-Plus trial.

**Table 1 ijerph-17-03728-t001:** Baseline characteristics of the participants according to baseline PA levels. Light = 0–1200 METs–min/weeks; moderate = 1200–1800 METs–min/week; vigorous= >2800 METs–min/week. Mean ± SD for age and body mass index (BMI). (%) for type 2 diabetes. depression. blood pressure. history of cancer. history of lung disease. smoking status. marital status and educational level. PA levels were compared with age and BMI with ANOVA test and with diabetes, depression, blood pressure, History of cancer, history of lung disease and smoking status. Marital status and educational level with a χ^2^ test. Significant *p* values in bold.

		PA											
		Total				Men				Women			
		Light	Moderate	Vigorous	*p-*Value	Light	Moderate	Vigorous	*p-*Value	Light	Moderate	Vigorous	*p-*Value
METs–min/week		**0–1200**	**1200–2800**	**>2800**		**0–1200**	**1200–2800**	**>2800**		**0**–**1200**	**1200–2800**	**>2800**	
*N*		2217	2206	2297		977	1079	1417		1240	1127	880	
Age (y)		64 (5)	65 (5)	65 (5)	**<0.001**	62 (5)	64 (5)	64 (5)	**<0.001**	66 (4)	66 (4)	66(4)	0.356
Body-mass index (kg/m^2^)		33.2 (3.5)	32.4 (3.3)	32.0 (3.1)	**<0.001**	33.0 (3.3)	32.1 (3.1)	31.9 (3.0)	**<0.001**	33.4 (3.6)	33.7 (3.5)	32.2 (3.3)	**<0.001**
Type 2 Diabetes at baseline (%)		29.4	26.1	26.3	**0.048**	30.6	28.8	28.5	0.579	28.4	23.5	22.7	**0.017**
Depression at baseline (%)		23.9	20.8	17.7	**<0.001**	12.4	13.1	10.9	0.248	32.9	28.1	28.5	**0.021**
Systolic blood pressure ≥140 mmHg (%)		44.4	47.7	48.6	**0.013**	46.3	51.5	53.3	**0.003**	43	44.1	41	0.382
Systolic blood pressure ≥90 mmHg (%)		18.5	18.9	19.1	0.865	24.8	23.9	22.2	0.325	13.6	14.1	14.1	0.907
History of cancer (%)		8.5	7.3	5.8	**0.002**	4	5.9	4.2	0.068	12.1	8.6	8.4	**0.004**
History of lung disease (%)		5.2	4.1	4.3	0.157	5.7	4.5	4.7	0.415	4.9	3.7	3.5	0.238
Smoking status (%)					**<0.001**				**<0.001**				**0.002**
	Current smoker	47.4	44.9	39.9		22.6	19.6	23.1		66.9	69	66.9	
	Former smoker	37.8	43.4	48.9		56.4	63.8	62.5		23.2	24	27.2	
	Never smoker	14.8	11.7	11.2		21	16.6	14.5		10	7	5.9	
Marital status (%)					**0.004**				0.150	10			0.234
	Married	74.7	75.8	78.9		82.8	84.7	86.5		68.3	67.2	66.8	
	Single	5.4	5	5.1		5.6	4.3	4		5.2	5.6	7	
	Divorced	9.1	7.6	7.1		8.7	7.6	6.3		9.4	7.5	8.4	
	Widowed/widower	10.9	11.7	8.8		2.9	3.4	3.3		17.1	19.7	17.8	
Educational level (%)					0.939				**0.006**				0.468
	Primary school or less	49.7	48.9	48.4		35.2	37.5	42		61.1	59.8	58.8	
	Secondary school	28.6	29.1	29.2		33.9	32.3	32		24.4	26.1	24.7	
	High school or university	21.7	22	22.4		30.9	30.2	26		14.4	14.1	16.6	

**Table 2 ijerph-17-03728-t002:** Baseline characteristics of the participants according to baseline BMI categories <30 kg/m^2^; 30–32.4 kg/m^2^; 32.5–34.9 kg/m^2^; ≥35 kg/m^2^. Mean ± SD for age and METs.min/week. (%) for type 2 diabetes, depression, blood pressure, history of cancer, history of lung disease, smoking status, marital status and educational level. BMI categories were compared with age and METs.min/week with ANOVA test and with diabetes, depression, blood pressure, history of cancer, history of lung disease, smoking status, marital status and educational level with a χ^2^ test. Significant *p* values in bold.

	BMI														
Total	Men	Women
kg/m^2^			**<30**	**30–32.4**	**32.5–34.9**	**>35**	***p-*** **Value**	**<30**	**30–32.4**	**32.5–34.9**	≥**35**	***p-*** **Value**	**<30**	**30–32.4**	**32.5–34.9**	≥**35**	***p-*** **Value**
*N*			1.747	1.846	1.517	1.610		959	1.032	780	702		788	814	737	908	
Age (SD)			65 (5)	65 (5)	65 (5)	65 (5)	0.093	64 (5)	64 (5)	63 (5)	63 (5)	**0.003**	66 (4)	67 (4)	66 (4)	66 (4)	0.352
	PA (Mets-min/week)	2724.6(2309.4)	2595.1 (2376.1)	2473.1 (2381.4)	2014.7 (2059.5)	**<0.001**	3108.9 (2524.7)	2944.7 (2667.0)	2852.5 (2711.9)	2369.6 (2377.2)	**<0.001**	2256.8 (1916.9)	2151.9 (1854.8)	2071.6 (1893.0)	1740.3 (1727.5)	**<0.001**
	Type 2 diabetes at baseline (%)	26.3	26.1	27.5	29.4	0.116	29.7	27.8	29	30.8	0.192	22.1	23.8	25.9	28.4	0.121
	Depression at baseline (%)	17.9	19.9	21.6	24	**<0.001**	10.5	11.8	13.4	12.7	0.306	26.8	30	30.4	32.7	0.068
	Systolic blood pressure ≥140 mmHg (%)	42.9	48.2	46	50.8	**<0.001**	47.3	50.5	51.4	55.1	**0.019**	37.4	45.3	40.3	47.4	**<0.001**
	Diastolic blood pressure ≥90 mmHg (%)	15.3	19.1	20.4	20.9	**<0.001**	19.7	23.6	25.5	26.2	**0.006**	9.9	13.5	15.1	16.7	**0.001**
	History of cancer (%)	7.6	6.7	7.2	7.5	0.736	6	4.3	5.1	3.1	**0.048**	9.5	9.7	9.4	10.8	0.749
	History of lung disease (%)	4.1	3.9	5	5.2	0.155	4.5	3.9	5.5	6.3	0.106	3.7	3.8	4.5	4.4	0.798
	Smoking status (%)					**<0.001**					0.120					**<0.001**
		Current smoker	42.6	40.5	44.7	48.8		23	21.5	20.6	22.2		66.4	64.7	70.2	69.3	
		Former smoker	42.1	46.8	43.4	41.2		57.5	62.8	63.8	60.9		23.4	26.5	21.9	26	
		Never smoker	15.3	12.7	11.9	10		19.5	15.7	15.7	16.9		10.2	8.9	7.9	4.8	
	Marital status (%)					0.078					0.113					0.702
		Married	77	77.8	77.5	73.6		84.5	85.1	86.7	83		67.7	68.4	67.8	66.3	
		Single	5	4.9	4.8	5.9		3.2	5	4.3	6		7.1	4.9	5.4	5.8	
		Divorced	8.7	7.6	7.3	8.1		8.6	7.2	5.9	7.6		8.7	8.1	8.7	8.5	
		Widowed/widower	9.5	9.7	10.4	12.6		3.7	2.7	3.1	3.4		16.5	18.6	18.1	19.5	
	Educational level (%)					**<0.001**					0.259					**0.003**
		Primary school or less	44.1	48.3	51.9	52.4		35.7	38.8	41	40.2		54.3	60.4	63.4	61.9	
		Secondary school	30.2	29.4	27.9	28.3		33	32.8	32.1	32.6		26.8	25.1	23.5	24.9	
		High school or university	25.8	22.3	20.2	19.3		31.4	28.5	26.9	27.2		18.9	14.5	13.2	13.2

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
