# Peer review of "The Effect of Physical Activity and High Body Mass Index on Health-Related Quality of Life in Individuals with Metabolic Syndrome"

_ijerph, 2020, doi:10.3390/ijerph17103728_

Round 1

Reviewer 1 Report

Thank you for your work.  The size and scope of this study is impressive and I look forward to reading the results of the randomized trial. 

Having such a large sample, all having metabolic syndrome, is a great contribution to the literature.  I have some minor concerns and some larger questions. 

-When you calculated the Met.min/week, you divided by 48 weeks/year.  Why not 52 weeks/year?

-As the randomized trial was meant to increase physical exercise with behavioral therapy and diet changes, it might be nice to state that the data in this analysis were collected before the intervention began.  I assume that is correct.

-You mention on line 403 that one of your covariates measured was diabetes, but on line 431, you state that you adjusted for type 2 diabetes.  Did you only ask about type 2 diabetes or did you ask about type 1 diabetes as well?  If you only asked about type 2 diabetes, please make that clear throughout. 

-The importance of this study is the fact that all 6720 had metabolic syndrome.  That being said, you don't mention that at all in your discussion.  You don't seem to compare your results to any other studies who used a similar (yet probably smaller) population.  This is a point of distinction and needs to be addressed in a much more visible way. 

-Due to your cut points, you had one group that was normal weight/overweight and the other three would be considered obese.  I would be careful making a statement such as "This suggests that both PA and obesity/overweight are determinants in HRQoL." as you don't have a group that is just made up of overweight individuals. Can you separate into a normal weight group and an overweight group and then see what happens with your analyses?

-You have a number of places where you spell out a word after you have abbreviated it or you abbreviate it after the first time you use it.  That needs to be consistent. 

-You switch between different tenses throughout the paper, which makes it hard to read.  Research has already taken place, so it needs to be in the past tense.

-You use both the term sex and gender.  Unless you are asking about gender identity, it should be sex. 

Reviewer 2 Report

Abstract

  • Please remove the “e” from the end of vigorous.
  • Remove the word increase on line 284. Increase is already implied because the authors state that there is a positive correlation.

Introduction

  • Lines 301-310: In addition to diseases being associated with obesity, a major obstacle to physical activity for adults with obesity, are impairments in movement and walking. This would be important to highlight for readers to understand how motor capacity relates to participation in physical activity (see the following papers:
    • Biomechanical gaitanalysis in obese men. Spyropoulos P, Pisciotta JC, Pavlou KN, Cairns MA, Simon SR. Arch Phys Med Rehabil. 1991 Dec;72(13):1065-70.
    • Gill, S. V., Hicks, G., Zhang, Y., Niu, J, Apovian, C. M., & White, D. K. (2017). The association of waist circumference with community walking ability in knee osteoarthritis: The osteoarthritis initiative. Osteoarthritis & Cartilage, 25: 60-66
    • The biomechanics of adiposity--structural and functional limitations of obesityand implications for movement. Hills AP, Hennig EM, Byrne NM, Steele JR. Obes Rev. 2002 Feb;3(1):35-43. Review.
    • Gill, S. V., Walsh, M. K., Pratt, J. A., Toosizadeh, N., Najafi, B., & Travison, T. G. (2016). Changes in spatio-temporal gait patterns during flat ground walking and obstacle crossing one year after bariatric surgery. Surgery for Obesity and Other Related Diseases, 12, 1080-5.).
    • Best practice & research clinical endocrinology & metabolism C Resteghini, S Cavalieri, D Galbiati, R Granata, S Alfieri, C Bergamini, Best Practice & Research Clinical Endocrinology & Metabolism 30, 1e13
    • Gill, S. V. & Narain, A. (2012) Quantifying the effects of body mass index on safety: Reliability of a video coding procedure and utility of a rhythmic walking task. Archives of Physical Medicine and Rehabilitation, 93, 728-730.
    • Forhan, M. & Gill, S. V. (2011, April). Cross border contributions to obesity research and interventions: A north american review of occupational therapy. Canadian Obesity Summit. Montreal, Canada.

Method

  • Were any corrections done to account for multiple comparisons?

Results

  • In which of the groups were the 82/122 people who reported engaging in moderate-intense exercise?
  • In Table 3, please also report the percent excess body mass lost for each group.
  • Please add error bars to Figure 2 so that readers can see whether overlaps occur more easily.

Discussion

  • Many of the findings hinge on comparing men and women. The authors address this very briefly in the discussion, but I think that this point deserves more than what was provided.
  •  

Reviewer 3 Report

The value of the article is, from the beginning, guaranteed by different projects and grants of relevance, at European level. Perhaps, not belonging to the specific research field of departure of the investigation, the conclusions are too little explicit and very generic. Perhaps this section should be expanded and made more specific.
It incorporates the necessary information guiding the reader to identify the basic content of the paper quickly and to determine its relevance. It is semantically self-sufficient. Explains the purpose and aims of the text presented.

The title synthesizes the main idea of the writing, it is explanatory by itself. In the same way it is concise and informative.

It presents the objective of the study, the main elements of the methodology and the findings and conclusions.

Current and relevant article.

As for the analysis and discussion, they are correctly presented, resuming the methodological procedures and conceptual guidelines.

Very adequate graphics, which complement the information in a detailed way.

Limitations and prospective are not explicit and should be proposed.

The article should be published, motivated by the joint quality of the article.

Round 2

Reviewer 2 Report

The authors have satisfactorily revised the manuscript. I think that it is ready to be accepted for publication.